# Interplay of opposing fate choices stalls oncogenic growth in murine skin epithelium

**Madeline Sandoval[1,2], Zhe Ying[1], Slobodan Beronja[1]***

[1]Division of Human Biology, Fred Hutchinson Cancer Research Center, Seattle, United States; [2]Molecular and Cellular Biology Graduate Program, University of Washington, Seattle, United States

**Abstract** Skin epithelium can accumulate a high burden of oncogenic mutations without morphological or functional consequences. To investigate the mechanism of oncogenic tolerance, we induced Hras[G12V] in single murine epidermal cells and followed them long term. We observed that Hras[G12V] promotes an early and transient clonal expansion driven by increased progenitor renewal that is replaced with an increase in progenitor differentiation leading to reduced growth. We attribute this dynamic effect to emergence of two populations within oncogenic clones: renewing progenitors along the edge and differentiating ones within the central core. As clone expansion is accompanied by progressive enlargement of the core and diminishment of the edge compartment, the intraclonal competition between the two populations results in stabilized oncogenic growth. To identify the molecular mechanism of Hras[G12V]-driven differentiation, we screened known Ras-effector in vivo and identified Rassf5 as a novel regulator of progenitor fate choice that is necessary and sufficient for oncogene-specific differentiation.

## Introduction

Adult human skin harbors potent oncogenic mutations without any functional or morphological consequences. A striking feature of this surprising tolerance is that mutant clones that persist in the tissue are both limited and uniform in size, irrespective of a mutation's mitogenic potential (*Martincorena et al., 2015*). In murine skin, gain-of-function mutant p53 clones grow extensively in naive epidermis, yet appear growth-inhibited when induced in a tissue premutagenized via acute exposure to UV. This suggests that the UV radiation introduced mutations that were able to compete with expanding p53 clones, limiting their size (*Murai et al., 2018*). This model of interclone growth inhibition may contribute to epithelial homeostasis in aged tissues, which contain a multitude of established mutations (*Bowling et al., 2019*; *Higa and DeGregori, 2019*).

There is also evidence that growth of oncogenic clones can be ameliorated in the naive epidermis. Activating mutation in β-catenin leads to ectopic hair follicle growths, which regress and are permanently eliminated (*Brown et al., 2017*). Epidermal activation of Pik3ca, the most commonly mutated oncogene in epithelial cancer, induces clones which are more proliferative than wild type (WT). However, they too are lost as the oncogenic progenitors are removed from the tissue due to an inability to self-renew (*Ying et al., 2018*). Because the oncogenic clones in these studies are eliminated, neither addresses how a single oncogenic clone may be tolerated to persist in the tissue long term. This is critical as the full complement of mutations in the aged epidermis likely arose through step-wise accumulation of discrete oncogenic mutations throughout the life of the tissue.

To answer this question, we use mouse skin and single-cell activation of oncogenic Hras (Hras[G12V]) which we previously demonstrated results in formation of persistent clones within the interfollicular epidermis (IFE) (*Beronja et al., 2013*). We follow Hras[G12V] single cells as they grow

*For correspondence:
beronja@fredhutch.org

Competing interests: The authors declare that no competing interests exist.

into stable clones and employ a novel cell fate identification (CFI) assay which allows us to discern temporal and spatial influences on progenitor cell renewal and differentiation. We demonstrate that expanding Hras$^{G12V}$ clones develop intraclonal heterogeneity that impacts progenitor cell fate choice in a manner that restricts their growth yet ensures their persistence and does not depend on additional mutations or interclonal competition. To identify the mechanism, we conducted a genetic screen of Ras effectors and identify Rassf5 as a novel candidate regulator of progenitor cell fate choice. Lastly, using gain-/loss-of-function studies in vivo we demonstrate that Rassf5 is necessary and sufficient for progenitor cell differentiation and inhibition of Hras$^{G12V}$-induced growth.

## Results

### Expression of activated Hras$^{G12V}$ promotes progenitor cell renewal in adult skin epithelium

In order to explore the immediate effect of single-cell activation of a potent oncogene on progenitor cell fate, we employed an *Hras$^{lox-WT-stop-lox-G12V}$* (*Hras$^{ll-G12V}$*); *Rosa26$^{mT/mG}$* (*R26$^{mT/mG}$*) mouse (*Figure 1A*; *Chen et al., 2009*; *Muzumdar et al., 2007*). We transduced the epidermis of mid-gestation embryos, using ultrasound-guided intraamniotic injection (*Beronja et al., 2013*; *Beronja et al., 2010*), with lentivirus containing inducible Cre-recombinase (LV-CreER) at clonal density (*Figure 1A*). Together, this model allowed for: (i) Tamoxifen-induced recombination in the adult epidermis, at a dose determined to result in sporadic single-cell activation events; (ii) expression of an oncogenic form of Hras (Hras$^{G12V}$) from its endogenous promoter, ensuring both physiological mRNA/protein levels and regulatory control; and (iii) stable labeling of activated cells by Cre-mediated replacement of membrane-associated (m) Tomato fluorescent protein with mGFP (*Figure 1A*). We induced recombination at postnatal day 19 (P19) and imaged tissues 48 hr later, allowing sufficient time for individual progenitor cells to divide and for the resulting two daughter cells to commit to either basal progenitor or suprabasal differentiated cell fate (*Figure 1B,C*; *Ying et al., 2018*). Using intravital two-photon microscopy (*Rompolas et al., 2013*), we were able to image through the full thickness of the epidermis, identify the differentiation state of each of the two daughter cells derived from a single activated cell at P21, and score every progenitor cell division that occurred as either symmetric renewal, asymmetric division, or symmetric differentiation (*Figure 1C,D*).

Analysis of control (*R26$^{mT/mG}$*) head skin epidermis showed that asymmetric divisions, once considered the predominant division type of epidermal progenitors, occurred in approximately 44% of cases, and that symmetric renewal was nearly perfectly balanced by symmetric differentiation (29% vs. 27%) (*Figure 1E*). In contrast, we observed a striking change in progenitor division type following expression of Hras$^{G12V}$, with a significant increase in symmetric renewal (48%) and a significant decrease in symmetric differentiation (10%) (*Figure 1E*). We used the observed frequency of cell division types in control and oncogene-expressing epidermis to calculate the rate of progenitor renewal, which measures the proportion of new daughter cells that maintain progenitor potential (*Ying et al., 2018*). Epidermal progenitors in the control tissue had a renewal rate of ~0.5 (*Figure 1F*). Such a rate is expected to maintain a stable population of basal progenitors long term and ensure neutral tissue growth. Hras$^{G12V}$-expressing progenitors showed higher renewal rate of 0.69, which implies that oncogenic Hras, unlike single-cell expression of activated Pik3ca (*Ying et al., 2018*), promotes progenitor cell renewal. It also suggests that in contrast to loss of cellular fitness we observed in oncogenic Pik3ca epidermis (*Ying et al., 2018*), Hras$^{G12V}$-expressing progenitors could support expansion and long-term maintenance of oncogenic clones. We tested the long-term persistence of oncogenic clones by quantifying the size of Hras$^{G12V}$ clones at 24 weeks (*Figure 1G, H*). At this point, Hras clones were significantly larger that WT clones but were smaller than anticipated based on the high renewal rate (0.69) at clone initiation. We first investigated apoptosis or senescence as a means to control clone growth but neither were significantly different between WT and oncogene expressing epidermis, (*Figure 1—figure supplement 1A,B*). This suggests that additional mechanisms of growth suppression are at play to curtail the expansion of Hras$^{G12V}$ progenitor cells.

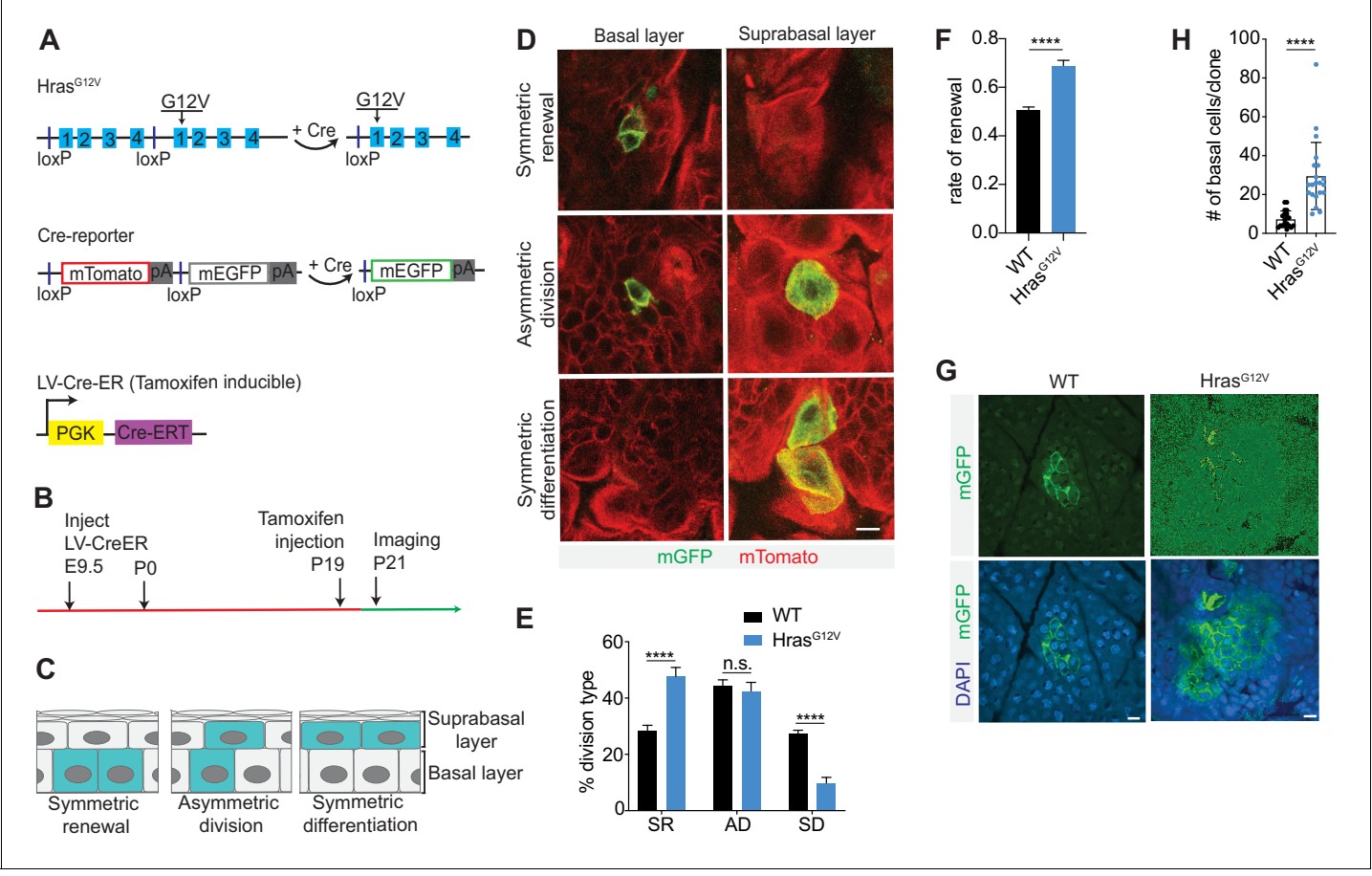

**Figure 1.** Hras[G12V] induces progenitor cell renewal in single cells. (A) Schematic of *Hras[ll-G12V]* and *R26[mT/mG]* mice. *Hras[ll-G12V]* mouse changes expression of wild-type (WT)-Hras allele to mutant-Hras allele upon addition of Cre. *R26[mT/mG]* mouse changes expression of mTomato to mGFP after Cre activation. CreER expression is induced by tamoxifen. (B) Schematic of single-cell induction and imaging experiment. (C) Epidermal progenitor cells can undergo three distinct division types. (D) Representative images of basal progenitor cell divisions in Cre-activated *R26[mT/mG]* epidermis captured using intravital imaging. Scale bar is 10 μm. (E) Quantification of division choices of single cells in adult epidermis. At least 75 cells were scored per animal. n ≥ 3 animals per genotype. (F) Rate of renewal significantly increases in single Hras[G12V] progenitor cells. At least 75 cells were scored per animal. n = 3 animals per genotype. (H) Quantification of basal cell numbers in WT and Hras[G12V] clones at 24 weeks. Each point represents one clone. Twenty total clones were scored from n ≥ 3 animals per genotype. (G) Representative images of basal cells of WT and Hras[G12V] clones at 24 weeks. Scale bar is 10 μm. For (E,F,H), the center line represents the mean; errors bars represent the s.d. Two-tailed Student's t-test was used. n.s. denotes p value > 0.05; **** denotes p value < 0.0001.

The online version of this article includes the following figure supplement(s) for figure 1:

**Figure supplement 1.** Hras G12V expression has no impact on apoptosis or senescence in skin epithelium.

## Development of an assay for quantification of progenitor cell fates in vivo

Next, we set to determine if high proliferation and renewal rates were sustained in Hras[G12V] clones over time. While proliferation could be measured using standard approaches, a simple and broadly accessible assay that can directly score progenitor cell fate choice was lacking. We recently employed an EdU/BrdU differentiation assay (*Ying et al., 2018*) that allowed us to analyze progenitor daughter cells as a population, and estimate tissue renewal rate based on their relative expression of differentiation marker Keratin 10 (K10). Here, we modified this approach to develop a CFI assay, which allows us to classify each progenitor cell division event as asymmetric, symmetric renewal, or differentiation. We first gave animals a pulse of EdU and 30 min later processed the tissue for standard confocal imaging (*Figure 2A*). We observed EdU+ labeling of single cells throughout the tissue that was restricted to the progenitors in the K10- basal layer (*Figure 2B*). We followed

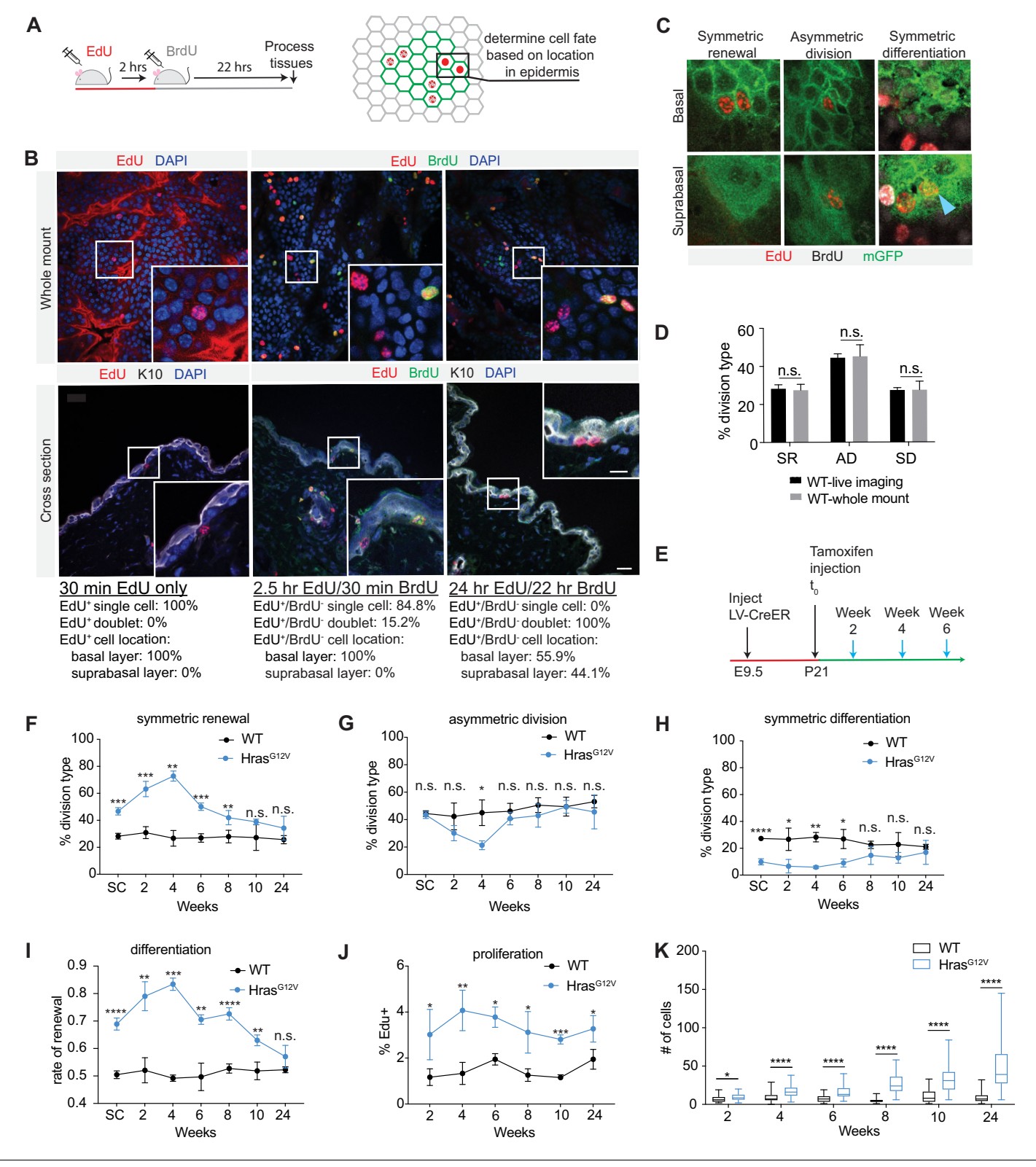

**Figure 2.** Cell fate identification (CFI) assay reveals dynamic changes in progenitor renewal in expanding Hras$^{G12V}$ clones. (**A**) Schematic of CFI assay. EdU-only cells are assessed for location in epidermis (basal or suprabasal layer) as indicator of cell fate choice. (**B**) Representative images of tissues stained for EdU and BrdU expression, 30 min, 2.5 hr, and 24 hr post-EdU injection. Cells at the 24 hr time point are fully committed to progenitor or differentiated fate. Scale bar is 25 μm. Inset scale bar is 10 μm. At least 2000 total cells were counted per animal. n ≥ 2 animals per time point. (**C**)

*Figure 2 continued on next page*

*Figure 2 continued*

Representative images of cell divisions as seen in CFI assay. Blue arrow marks symmetric differentiation division. (D) Cell fate choices scored by CFI assay are not significantly different from cell fate choices scored via intravital imaging. At least 50 cells were scored per animal. n ≥ 3 animals per condition. (E) Schematic of clone activation. Tissues were processed in 2-week intervals following tamoxifen injection (blue arrows). (F–H) Division choices of progenitor cells in activated clones. At least 30 divisions were counted per animal. n ≥ 3 animals per condition. (I) Rate of renewal in Hras$^{G12V}$ and WT clones initially differs significantly, but as Hras$^{G12V}$ clones reach 24 weeks, renewal rates drop to near-homeostatic levels. More than 30 total cells were counted per animal. n ≥ 3 animals per condition. (J) EdU-incorporation over 2 hr in WT and Hras$^{G12V}$ epidermis. At least 300 cells were counted per animal. n = 3 animals per condition. (K) Hras$^{G12V}$ clones expand significantly over time. At least 25 clones were counted per animal. n ≥ 3 animals per condition. For (F–K), the center point represents the mean; errors bars represent the s.d. Two-tailed Student's t-test was used. n.s. denotes p value > 0.05; * denotes p value < 0.05; ** denotes p value < 0.01; *** denotes p value < 0.001; **** denotes p value < 0.0001.

the EdU pulse with administration of BrdU 2 hr later (*Figure 2A*). This resulted in the appearance of sporadic EdU$^+$/BrdU$^+$ single cells and a population of EdU$^+$/BrdU$^-$ cells, in which ~ 15% of cells have already divided but not differentiated (*Figure 2B*). By 24 hr all EdU$^+$/BrdU$^-$ cells were found as doublets and in both basal (K10$^-$) and differentiated K10$^+$ suprabasal layers (*Figure 2B*). These EdU$^+$-only cells represent the daughters of progenitor cells that have exited the S-phase within a defined period (0–2 hr), and have been given sufficient time to show their differentiation state. By keeping the epidermis intact through processing and immunofluorescence staining, we are able to image every EdU$^+$-only doublet in tissue whole-mounts and assign daughter cell fates based on cellular morphology (cuboidal vs. flat) and location in the tissue (basal vs. suprabasal; *Figure 2C*).

We next compared the frequency of cell division types in control epidermis as measured by CFI with that obtained using intravital two-photon imaging of LV-CreER-transduced single cells (*Figures 1E* and *2D*). We observed no significant differences between the two assays, suggesting that in CFI we have a valid and direct method for quantifying cell fate decision within the adult murine epidermis.

## Epidermal expression of Hras$^{G12V}$ results in dynamic changes in progenitor cell fate choice

To investigate how proliferation and cell fate choice may evolve in Hras$^{G12V}$ cells over time, and account for relatively restricted growth of oncogene-expressing clones (*Figure 1G,H*), we measured their proliferation and cell fate rates. We lineage traced control and Hras$^{G12V}$ clones initiated from a single cell at P21 and collected tissues at 2-week intervals (*Figure 2E*). We observed that oncogenic clones undergo more symmetric renewal initially, but by 8–10 weeks of growth the proportion of symmetric renewal divisions decreased and returned to rates observed in WT clones (*Figure 2F*). Early increase in symmetric renewal was accompanied by reduction in the rates of symmetric differentiation and asymmetric divisions, which also returned to WT levels by 8–10 weeks (*Figure 2G,H*). From the observed frequency of cell division types, we calculated the rate of progenitor renewal. The WT progenitors maintained a renewal rate of ~0.5 throughout the duration of the study, consistent with a neutral growth potential of a homeostatic tissue (*Figure 2I*). In contrast, the Hras$^{G12V}$-expressing progenitors showed an increase in renewal rate to ~0.70–0.85 in the initial 8 weeks, but then fell to ~0.55 over the next several weeks as differentiating and asymmetric cell divisions increased (*Figure 2I*).

We also measured proliferation rates in control and Hras$^{G12V}$ clones over the same period, and observed that while oncogene expressing cells had a higher rate of EdU incorporation than WT, the rates were largely stable over 24 weeks (*Figure 2J*). Together, this suggests that clonal expansion driven by a single oncogenic lesion is dependent on the cellular mechanisms of progenitor proliferation and renewal, with fluctuating rates of cell fate choice being the dominant determinant of growth dynamics (*Figure 2K*).

## Emerging heterogeneity in expanding Hras$^{G12V}$ clones impacts progenitor renewal behavior

Intrigued by the observed switch from highly renewing to more balanced cell fate choices we revisited oncogene-expressing clones early (weeks 2 and 4) and late (weeks 10 and 24) following single-cell Hras$^{G12V}$ activation (*Figure 3*). We used confocal microscopy of epidermal whole mounts to observe individual mGFP+ clones, and image processing to distinctly visualize their progenitor

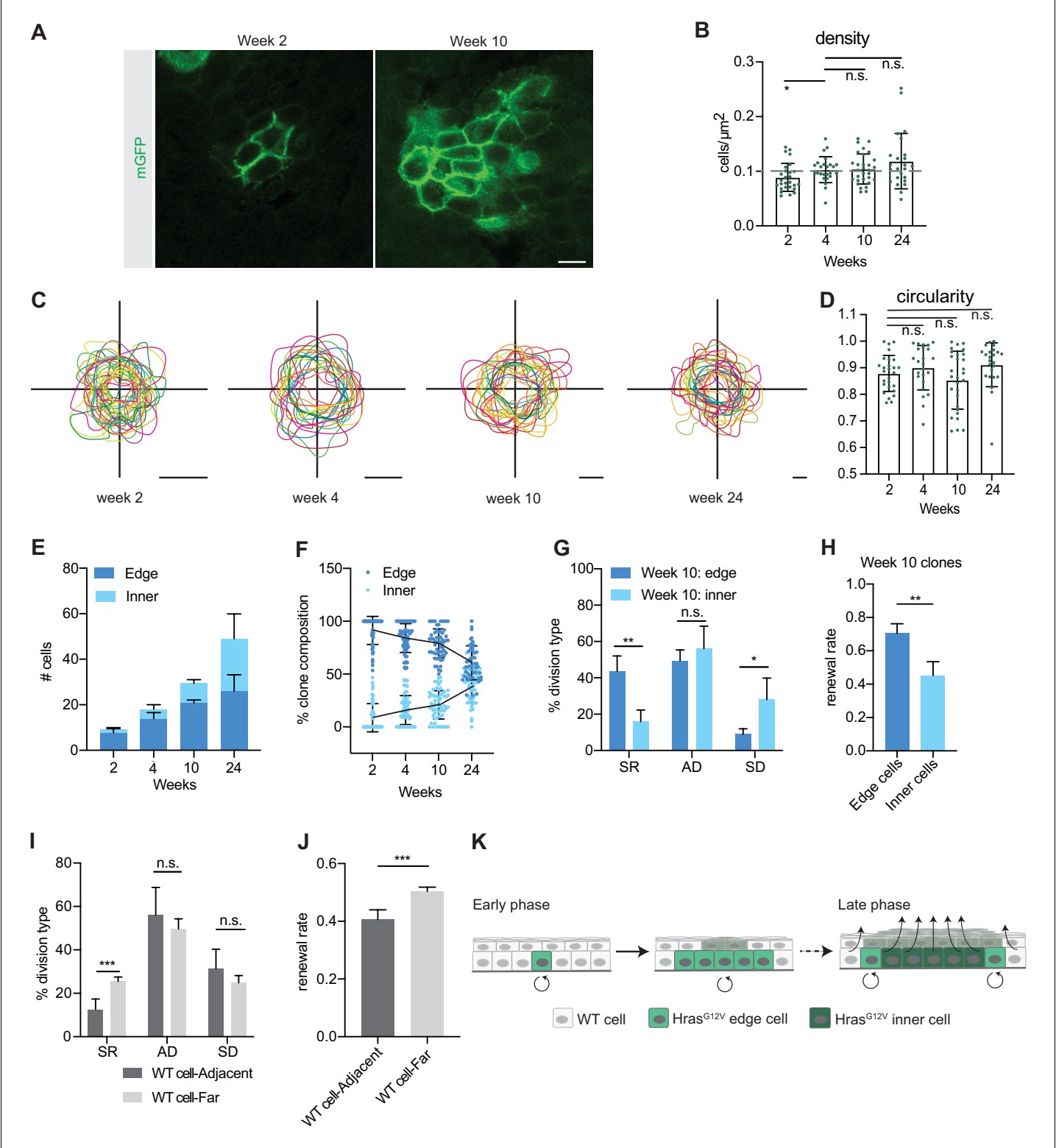

**Figure 3.** Hras[G12V] clones develop intraclone heterogeneity over time. (**A**) Representative images of basal cell population of Hras[G12V] clones at weeks 2 and 10. Scale bar is 10 µm. (**B**) Density of individual Hras[G12V] clones. Dashed gray line represents average density in wild-type (WT) clones. At least 25 clones were counted per animal. n ≥ 2 animals per timepoint. (**C**) Outlines of individual Hras[G12V] clones show a circular morphology. Scale bars are 25 µm. (**D**) Quantification of circularity of individual Hras[G12V] clones. At least 25 clones were counted per animal. n ≥ 3 animals per timepoint. (**E**) Quantification of edge cells and inner cells in Hras[G12V] clones. At least 52 clones were counted per animal. n ≥ 3 animals per time point. (**F**) Proportion

*Figure 3 continued on next page*

*Figure 3 continued*

of edge cells in Hras$^{G12V}$ clones decreases as the clone expands. At least 52 clones were counted per animal. n ≥ 3 animals per timepoint. (**G**) Edge cells undergo significantly more renewing divisions and significantly fewer differentiating divisions compared to inner cells in week 10 Hras$^{G12V}$ clones. A total of 147 divisions were counted. n = 4 animals. (**H**) Rate of renewal in edge cells and inner cells of week 10 Hras$^{G12V}$ clones. A total of 147 divisions were counted from four animals. (**I**) WT cells adjacent to an Hras$^{G12V}$ clone undergo significantly fewer renewing divisions compared to a non-adjacent WT cell. A total of 160 divisions were counted. n = 5 animals. (**J**) The renewal rate of WT cells adjacent to an Hras$^{G12V}$ clone is significantly lower compared to WT cells not located adjacent to an Hras$^{G12V}$ clone. A total of 160 divisions were counted. n = 5 animals. (**K**) Early phase Hras$^{G12V}$ clones expand. In contrast, the inner region of late phase Hras$^{G12V}$ clones undergo high rates of differentiation, but the clone does not collapse because the edge cells are more renewing and compensate for the inner cells. In response to the high rate of renewal in oncogenic edge cells, the neighboring WT cells undergo a high rate of differentiation. For (**B,D**) each point represents a single clone. For (**B, D–J**); the center point represents the mean; errors bars represent the s.d. Two-tailed Student's t-test was used. n.s. denotes p value > 0.05; * denotes p value < 0.05; ** denotes p value < 0.01; *** denotes p value < 0.001.

The online version of this article includes the following figure supplement(s) for figure 3:

**Figure supplement 1.** Circularity and cell fate dynamics of wild-type (WT) epidermal clones.

compartment (*Figure 3A*). Based on evidence that cell packing can have profound effect on delamination (*Lee et al., 1998*; *Poumay and Pittelkow, 1995*), we first compared clonal cell density between early and late Hras$^{G12V}$-expressing cells (*Figure 3B*). We observed that cell density increases between two and four weeks but then remains stable as highly renewing clones transition to more differentiating clones. This lack of temporal correlation suggested that cell density does not drive the switch in progenitor renewal.

We noted that the shape of expanding Hras$^{G12V}$ clones appeared to be relatively round, irrespective of their age and size (*Figure 3C*). Indeed, this was confirmed through analysis of Hras$^{G12V}$ clone circularity (*Figure 3D*) and further implied that, as clones expand, their area increases at a greater rate than their circumference. This suggested that time-dependent increase in clone size may be accompanied by a change in its overall composition in respect to the ratio of 'edge' cells, found along the circumference of the clone and in contact with WT neighbors, to 'inner' cells, found in the center of the clone and in contact with other Hras$^{G12V}$-expressing progenitors. To test this, we quantified the number of edge and inner cells in Hras$^{G12V}$ clones and observed that while both populations enlarged in absolute terms (*Figure 3E*), the relative abundance of inner cells increased from 14 to 45% and edge cells decreased from 86 to 55% over time (*Figure 3F*). Therefore, as Hras$^{G12V}$ clones expand, their composition changes in a potentially profound way.

We hypothesized that the heterotypic (WT/Hras$^{G12V}$) environment of the edge cells in Hras$^{G12V}$ clones could influence them to behave differently than the homotypic (Hras$^{G12V}$/Hras$^{G12V}$) inner cells. Using CFI assay, we analyzed 10-week-old Hras$^{G12V}$-clones and observed that although the rate of homeostatic asymmetric divisions was similar between edge and inner cells, they significantly differed in the ratio of growth regulating symmetric fate choices (*Figure 3G*). Although edge cells underwent significantly more symmetric renewing divisions, inner daughter cells more often chose differentiated cell fates. From observed division types, we extrapolated the renewal rate and established that edge cell population maintains progenitor renewal rate consistent with growth expansion (0.69) while inner population is below the 0.5 threshold required for stasis, and therefore in a state of progenitor loss (*Figure 3H*).

We did the same analysis in early- and late-stage WT clones and observed that while they too remain circular over time, their small size (~3 progenitor cells) prevents formation of distinct inner and edge cell compartments (*Figure 3—figure supplement 1A–C*). To approximate cell dynamics that may exist within a large lentivirus-transduced WT clone, we used multi-clonally derived fields of CreER-activated mGFP cells. We did not detect a significant change in division choices or the renewal rate in cells at the edge or the inner portion of the activated WT areas (*Figure 3—figure supplement 1D,E*). We also analyzed the behavior of WT cells adjacent to late-stage Hras$^{G12V}$ clones. We observed that these cells had a lower frequency of symmetric renewing divisions, as compared to WT cells not adjacent to Hras$^{G12V}$ clones (*Figure 3I*). This change was reflected by a significantly lower renewal rate (0.41) compared to distant WT cells (0.50) (*Figure 3J*).

Taken together, our data suggest a new model for how growth of oncogene-expressing cells becomes restricted to ensure normal function despite accumulation of cancer driving genes. According to it, a single Hras$^{G12V}$ cell initially undergoes a rapid expansion, but after several weeks, a shift

in renewal driven by intraclone heterogeneity decreases the rate of clone expansion until a point of near homeostasis. Even though the inner core of the clone is pro-differentiation, the clone does not collapse because it is maintained by the pro-renewal edge. Moreover, this pro-renewal edge is further supported by the reduced fitness of clone-adjacent WT cells that increase their rate of differentiation (*Figure 3K*). Our model also suggests the existence of a currently unknown molecular mechanism that regulates dynamic cell fate choice. Such a mechanism could explain how tissues can remain phenotypically normal while maintaining a high mutation burden throughout an individual's life.

## Functional screen identifies Hras effector Rassf5 as a mediator of cell fate choice

We have previously shown that developing epidermis is a powerful system to conduct large-scale genetic screens and identify physiological regulators of distinct cellular processes, including progenitor renewal and differentiation (*Beronja et al., 2010*; *Cai et al., 2020*; *Ying et al., 2018*). We next set out to test if the dynamic impact of Hras$^{G12V}$ on progenitor renewal in adult skin can be recapitulated in oncogene-expressing embryonic epidermis. We first injected low titer LV-Cre (*Beronja et al., 2010*), encoding a constitutively active Cre recombinase, into control (*R26$^{YFP}$*) and test (*Hras$^{ll-G12V}$*; *R26$^{YFP}$*) embryos at E9.5 (*Figure 4A*). At E18.5, we confirmed that epidermis was transduced at clonal density (*Figure 4A,B*), and used EdU/BrdU differentiation assay to quantify renewal rate in YFP$^+$ clones (*Ying et al., 2018*). We observed that progenitor renewal rates were high (~0.75), and not significantly different between control and test clones (*Figure 4C*). We next transduced mice with high titer LV-Cre, which resulted in broad areas of Hras$^{G12V}$ activation by E18.5 (*Figure 4A,B*). We quantified progenitor renewal under large field expression of Hras$^{G12V}$ and discovered a significant rate reduction (0.76 vs. 0.57; p value < 0.001; *Figure 4C*). We also used K14-Cre transgenic mice to generate E18.5 epidermis with uniform field activation of Hras$^{G12V}$ (*Figure 4A,B*), and observed that the tissue's progenitor renewal rate (0.56) showed no difference between LV-Cre- and K14-Cre-induced fields (*Figure 4C*). Our data imply that a field of progenitor cells that ubiquitously express Hras$^{G12V}$ more often gives rise to differentiated daughter cells, and this correlates with the changes we see develop over time in adult Hras$^{G12V}$ clones. They also suggest that we may be able to use our E18.5 screening platform to uncover the mechanism of how activated Hras$^{G12V}$ induces differentiation.

With a goal of identifying immediate and direct events downstream of a powerful oncogenic signal, we generated a library of lentiviral constructs for expression of short hairpin RNAs (shRNAs) targeting 28 known effectors of Ras signaling (*Figure 4D*). We combined 4–5 shRNAs targeting each gene into a single pool of 135 constructs, titrated to result in MOI ≤ 1, and co-injected it along with high-titer LV-Cre into control (*R26$^{YFP}$*) and test (*Hras$^{ll-G12V}$*; *R26$^{YFP}$*) E9.5 epidermis, using ultrasound-guided in utero microinjection (*Figure 4D*). We collected the epidermis at E18.5, isolated the keratinocytes, and separated them into basal ($\alpha_6$ Itg$^{high}$) and suprabasal ($\alpha_6$ Itg$^{low}$) populations. We reasoned that if an shRNA is enriched in the basal layer, it would imply that the corresponding gene was a negative regulator of progenitor renewal, while suprabasal enrichment would be consistent with depletion of a renewal promoter. We quantified relative abundance of shRNAs in two populations using BWA and quantified significant basal/suprabasal enrichment using DeSeq2 (*Love et al., 2014*; *Ying et al., 2018*). Genes were scored as screen hits if at least two shRNAs showed significant and consistent enrichment, as before (*Beronja et al., 2013*).

Our screen in WT epidermis did not identify any Hras effector enrichment in basal or suprabasal cells (*Figure 4E*), suggesting that in normal development Hras signaling is not a significant regulator of progenitor cell fate choice. In contrast, the screen in Hras$^{G12V}$-expressing epidermis identified shRNAs targeting Rassf5 and Pik3ca as significantly enriched in basal cells (*Figure 4F*), consistent with their putative role as negative regulators of Hras$^{G12V}$-mediated progenitor renewal. The screen also identified shRNAs targeting Rassf4, Rgl2, and Rgs14 as significantly enriched in suprabasal cells (*Figure 4F*, *Supplementary file 1*), suggesting that these genes may positively regulate renewal in Hras$^{G12V}$ epidermis. We were encouraged by the observed enrichment of Pik3ca shRNAs in basal progenitors, as our lab has previously shown that activated Pik3ca can drive progenitor differentiation in the epidermis (*Ying et al., 2018*). We next set out to test our top hit, Rassf5, as a candidate mediator of oncogene-induced differentiation.

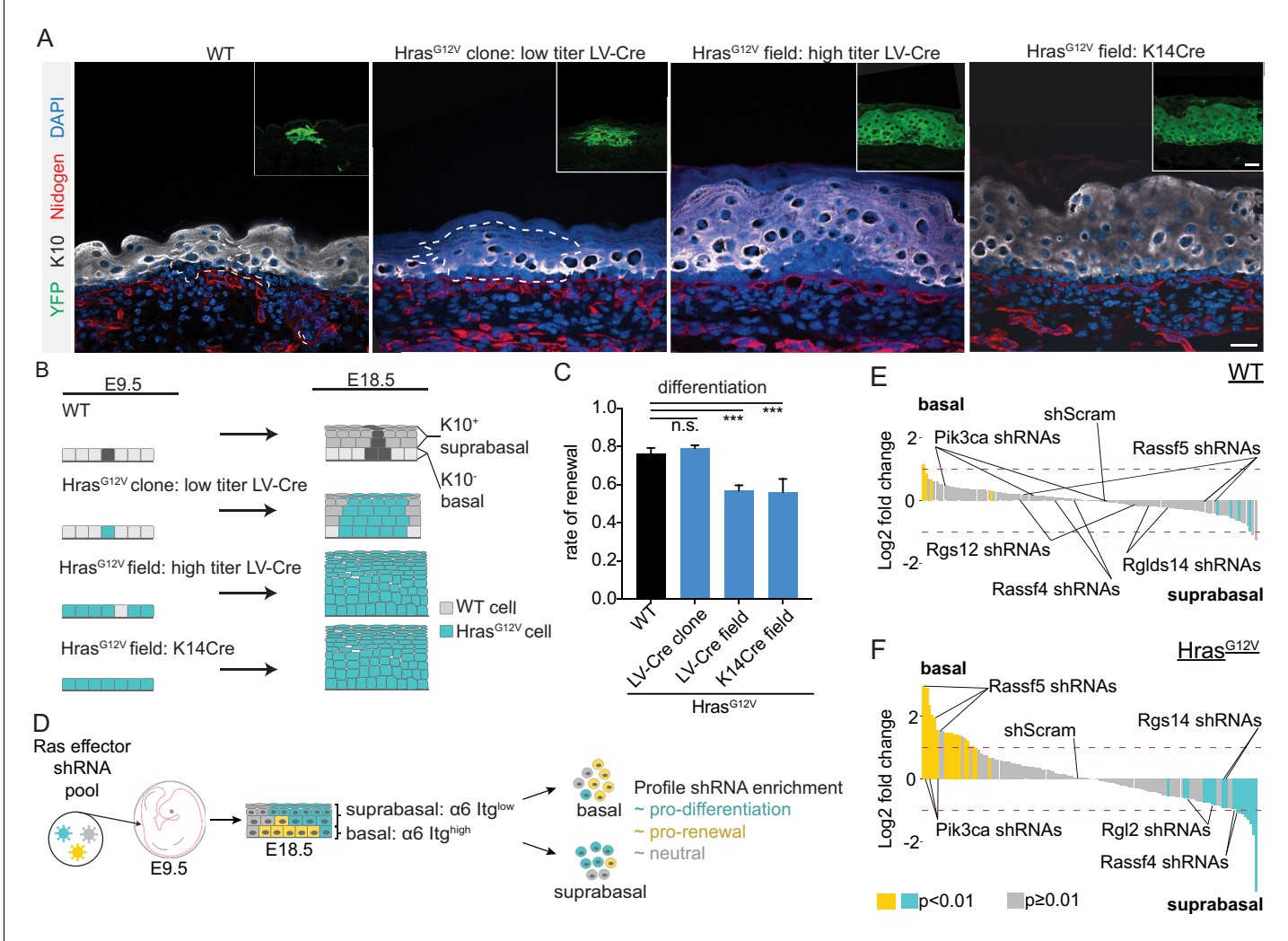

**Figure 4.** Hras^G12V-induced differentiation is replicated in E18.5 epidermis and serves as the basis for in vivo genetic screen. (**A**) Representative images of clonal and field activation of Hras^G12V in developing epidermis. Scale bar is 25 μm. Inset scale bar is 25 μm. (**B**) Hras^G12V expression can be induced with LV-Cre (high or low titer) or K14-Cre. Changes in Hras^G12V expression yield differences in tissue hyperplasia. (**C**) Rate of renewal decreases in tissues which broadly express Hras^G12V (LV-Cre field and K14Cre field). At least 75 cells were counted from each animal per genotype. n ≥ 3 animals. (**D**) Diagram of shRNA screen methodology. A lentiviral pool targeting Ras effectors was injected into E9.5 embryos. The epidermis was harvested at E18.5, digested into single cells and separated by FACS based on low or high expression of α6 Itg. Each population was profiled for shRNA enrichment. (**E,F**) Needle plots of shRNA's enriched in basal layer compared to suprabasal layer in wild-type (WT) and Hras^G12V epidermis. shRNAs that are enriched in basal layer (α6 Itg^high population) relative to suprabasal layer (α6 Itg^low population) inhibit differentiation. The indicated shRNAs are candidate (Rassf5) or validated (Pik3ca) regulators of Hras^G12V-induced differentiation in addition to potential positive regulators of renewal in Hras^G12V epidermis (Rassf4, Rgl2, Rgs14). n ≥ 3 animals per genotype. For (**C**), the center point represents the mean; errors bars represent the s.d. Two-tailed Student's t-test was used. n.s. denotes p value > 0.5, *** denotes p value < 0.001.

## Rassf5 mediates Hras-dependent reduction in progenitor cell renewal

Rassf5, also known as Nore1a, is a Ras effector containing a Ras Association (RA) domain which directly binds to activated Ras (*Vavvas et al., 1998*). Like other RA domain family (RASSF) members, Rassf5 lacks enzymatic activity and is often epigenetically silenced in cancer (*Donninger et al., 2016*). Notably, Rassf5 is methylated in the majority of esophageal and head and neck squamous cell carcinomas (*Guo et al., 2015*; *Steinmann et al., 2009*) and has been shown to inhibit Ras growth via senescence (*Donninger et al., 2015*) and apoptosis (*Elmetwali et al., 2016*; *Park et al., 2010*).

To discern if Rassf5 also regulates progenitor cell fate choice and is critical to Hras$^{G12V}$-mediated differentiation, we first quantified its expression in Hras$^{G12V}$ epidermis at E18.5 and discovered that both its transcript and protein were elevated relative to WT control, indicating that activated Hras$^{G12V}$ can promote Rassf5 expression (*Figure 5A,B*). We next identified three independent shRNAs which depleted Rassf5 mRNA by ~70% in keratinocytes (*Figure 5—figure supplement 1A*), and generated lentiviral constructs for concomitant expression of Rassf5 shRNAs and Cre (LV-Cre-sh*Rassf5*; *Figure 5C*). We introduced these into *Hras$^{ll-G12V}$*; *R26$^{mT/mG}$* epidermis at high MOI, isolated mGFP+ cells by FACS and profiled them for mRNA and protein levels to observe their significant potential to counter Hras$^{G1V}$-driven Rassf5 expression in vivo (*Figure 5A,B*).

To independently validate our screen results and test if Rassf5 is necessary for increased differentiation under conditions of field activation of Hras$^{G12V}$, we transduced control and *Hras$^{ll-G12V}$*; *R26$^{YFP}$* epidermis with high-titer LV-Cre-sh*Rassf5*, collected broadly infected tissue at E18.5 and performed EdU/BrdU differentiation assay (*Figure 5D*). Our data show that Rassf5 depletion in large Hras$^{G12V}$-expressing fields leads to a significant increase in progenitor renewal rate, with three independent shRNAs demonstrating the same effect. This signifies that Rassf5 is necessary for Hras$^{G12V}$-induced progenitor cell differentiation in embryonic tissues. We also observe increased renewal in control epidermis with one out of three shRNAs we used, which suggests that Rassf5 may have a more general ability to regulate progenitor cell renewal (*Figure 5D*).

To test if Rassf5 is sufficient to promote progenitor cell differentiation in embryonic tissue, we focused on small Hras$^{G12V}$ clones that we demonstrated do not show increased differentiation (*Figure 4C*). We generated a lentivirus to simultaneously express Cre recombinase and Rassf5 open-reading frame (ORF) fused to a tdTomato (LV-Cre-*Rassf5*-tdT), allowing for clonal oncogene activation and visualization of transgene expression (*Figure 5E*). We transduced Hras$^{G12V}$ epidermis with similar low titers of LV-Cre and LV-Cre-*Rassf5*-tdT, and readily observed numerous clones at E18.5. We noted a dramatic difference in their shape, with Rassf5-expressing clones appearing more triangular in cross-section (*Figure 5F*), consistent with either a significant loss of basal progenitors or expansion of the suprabasal differentiated cell compartment. Whole mount imaging further implied that Rassf5-expressing Hras$^{G12V}$ clones were composed of fewer basal and relatively more suprabasal cells (*Figure 5G*). We analyzed clone composition and observed that Rassf5 promotes significant reduction in basal cell numbers and significant increase in the suprabasal to basal surface area (*Figure 5H,I*). Together this suggests that Rassf5 is sufficient to promote progenitor cell differentiation in Hras$^{G12V}$-expressing embryonic epidermis.

We next sought to investigate whether Rassf5 can also regulate progenitor renewal in adult epidermis. We first analyzed Hras$^{G12V}$ epidermal clones where Rassf5 was depleted with two independent shRNAs. Focusing on the difference in renewal rate between the edge and inner clone compartments (*Figure 3H*), we observed that depletion of Rassf5 in adult Hras$^{G12V}$ epidermis specifically suppressed progenitor differentiation of inner cells (*Figure 5J*). As a result, the overall progenitor renewal rate was significantly elevated in Hras$^{G12V}$/shRassf5 tissues (*Figure 5K*). We also measured rates of apoptosis and senescence in adult Hras$^{G12V}$/shRassf5 epidermis and found neither to be significantly affected (*Figure 5—figure supplement 1B,C*). These observations suggest that Rassf5 is a negative regulator of progenitor renewal in the inner core of Hras$^{G12V}$ clones in the adult epidermis. To test next for the sufficiency of Rassf5 in promoting increased differentiation in adult Hras$^{G12V}$ epidermis, we utilized a system to overexpress Rassf5 in Hras$^{G12V}$-activated adult clones. We co-injected animals with a high-titer LV-TRE-*Rassf5*-tdT mixed with a low-titer LV-rtTA-T2A-CreER, thus generating a tissue with ubiquitous LV-TRE-*Rassf5*-tdT and sporadic LV-rtTA-T2A-CreER transduction (*Figure 5L,M*). We induced Hras$^{G12V}$ expression in single cells via administration of Tamoxifen at P21, which was followed by Doxycycline-mediated activation of Rassf5-tdT expression in cells co-transduced with both viruses at P24, and tissue harvesting at P27 (*Figure 5M*). Our analyses showed that resultant early clones expressing Hras$^{G12V}$ and Rassf5 had a significant reduction in their basal cell population compared to Hras$^{G12V}$ clones alone (*Figure 5O*). Moreover, we observed a significant increase in the number of Hras$^{G12V}$, Rassf5 clones without any basal cells (*Figure 5O*), consistent with Rassf5 inducing differentiation in adult Hras$^{G12V}$ clones. Together, our data indicate that Rassf5 is both necessary and sufficient positive regulator of Hras$^{G12V}$-mediated differentiation in adult as well as embryonic epidermis.

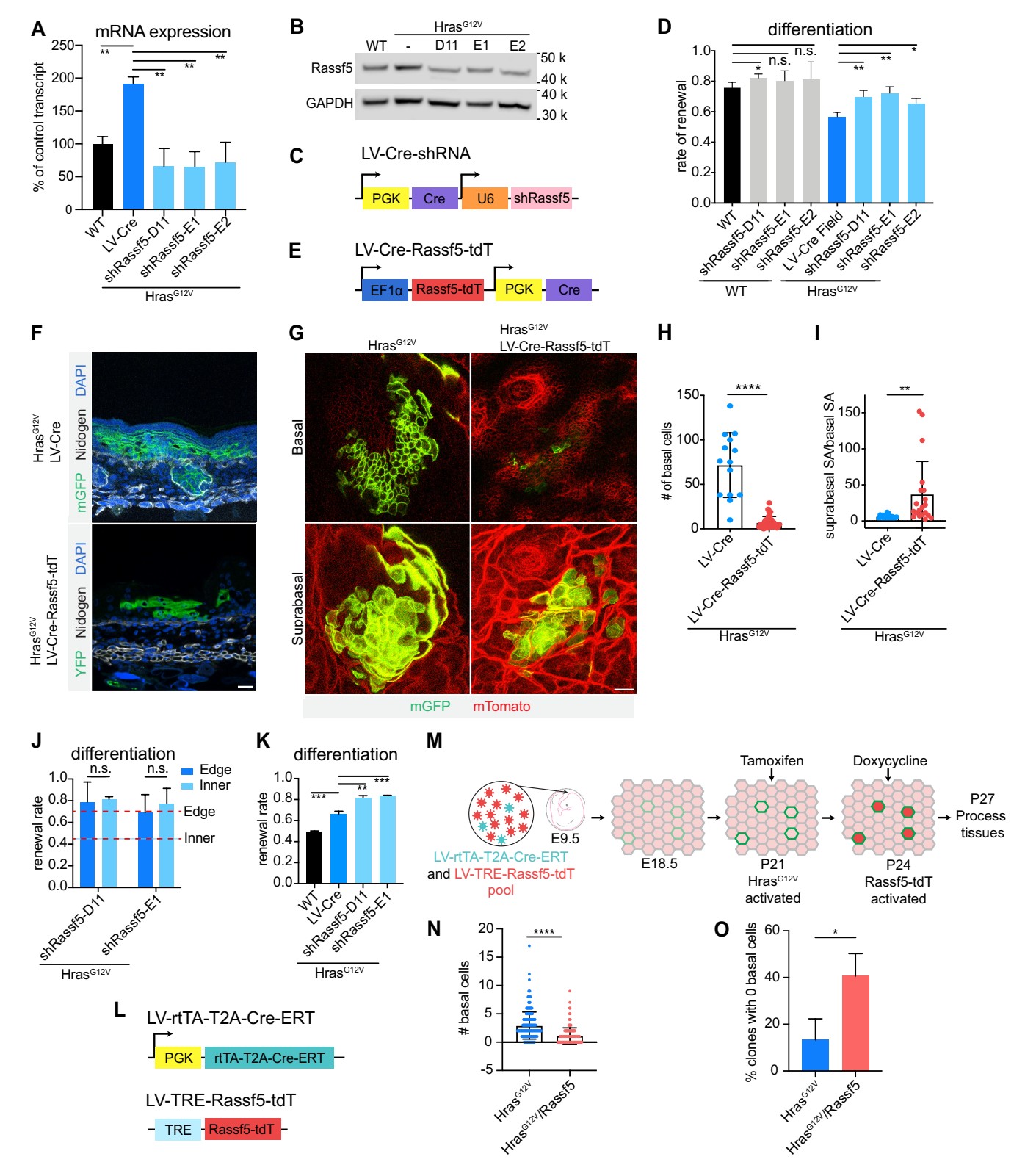

**Figure 5.** Rassf5 is a necessary and sufficient driver of Hras$^{G12V}$-induced differentiation. (**A**) Rassf5 mRNA is increased in isolated Hras$^{G12V}$ E18.5 basal cells. shRNAs targeting Rassf5 efficiently knock down transcript expression. n = 3 animals per genotype. (**B**) Immunoblot of Rassf5 expression in Hras$^{G12V}$ E18.5 epidermis. (**C**) LV-Cre-shRassf5 construct for simultaneous knock down of Rassf5 and induction of Hras$^{G12V}$ expression. (**D**) Quantification of differentiation using EdU-BrdU pulse/chase differentiation assay. Depletion of Rassf5 promotes renewal in Hras$^{G12V}$ epidermis. At least 75 cells were

*Figure 5 continued on next page*

*Figure 5 continued*

counted per animal. n ≥ 3 animals per genotype. (**E**) Diagram of construct containing Rassf5 open-reading frame (ORF) fused with tdTomato (Rassf5-tdT) and co-expressing Cre. (**F**) Representative images of Hras$^{G12V}$ clone and Hras$^{G12V}$/Rassf5-tdT clone in E18.5 epidermis. Overexpression of Rassf5 yields clones with reduced basal compartment and extensive suprabasal compartment. Scale bar is 25 μm. (**G**) Whole mount images of Hras$^{G12V}$ clone and Hras$^{G12V}$/Rassf5-tdT clone in E18.5 epidermis. Scale bar is 25 μm. (**H**) Quantification of basal cell numbers in E18.5 Hras$^{G12V}$ and Hras$^{G12V}$/Rassf5-tdT clones. Twenty-two clones were counted in total. n = 3 animals per genotype. (**I**) Quantification of the ratio of suprabasal surface area to basal surface area of Hras$^{G12V}$ and Hras$^{G12V}$/Rassf5-tdT clones in E18.5 epidermis. Twenty-two clones were counted in total. n = 3 animals per genotype. (**J**) Difference between the renewal rate of edge and inner cells is lost in Hras$^{G12V}$/shRassf5 clones. Animals are >10 weeks old. Red dash lines represent renewal rate of edge and inner cells from week 10 clones. At least 120 cells were counted per genotype. n = 3 animals. (**K**) Knock down of Rassf5 in adult Hras$^{G12V}$ epidermis significantly increases renewal rate. At least 75 cells were quantified per animal. n ≥ 3 animals per genotype. (**L**) LV-rtTA-T2A-CreER and LV-TRE-Rassf5-tdT constructs for inducible expression of Hras$^{G12V}$ and Rassf5. (**M**) Schematic of Rassf5 overexpression in adult Hras$^{G12V}$ clones. Developing embryos were broadly transduced with LV-rtTA-T2A-CreER and sporadically transduced with LV-TRE-Rassf5-tdT. At P21, tamoxifen injection induced expression of Hras$^{G12V}$, and at P24, doxycycline injection induced expression of Rassf5. Tissues were processed at P27. (**N**) Clones expressing Hras$^{G12V}$/Rassf5 have a reduced basal cell population compared to clones expressing Hras$^{G12V}$. n = 3 animals per condition. A total of 200 clones per condition were counted. (**O**) There are proportionally fewer Hras$^{G12V}$ clones composed of only differentiated cells compared to Hras$^{G12V}$/Rassf5. At least 59 clones were counted per animal. n = 3 animals per condition. For (**H,I,N**), each dot represents an individual clone. For (**A,D,H–K,N, O**), the center line represents the mean and errors bars the s.d. Two-tailed Student's t-tests were used. n.s. denotes p value > 0.05. * denotes p value < 0.05; ** denotes p value < 0.01*** denotes p value < 0.001; **** denotes p value < 0.0001.

The online version of this article includes the following figure supplement(s) for figure 5:

**Figure supplement 1.** Rassf5 depletion does not induce apoptosis or senescence in Hras G12V epidermal clones.

## Discussion

With CFI, a modified version of our EdU/BrdU pulse-chase assay (*Ying et al., 2018*) we have established a quantitative method that identifies individual progenitor cell fates that occur over a span of 24 hr in the adult epidermis. It is characterized by several features which should make it a standard in the field: (i) it quantifies progenitor renewal and differentiation rates in vivo—by including net effects of direct and indirect determinants such as spindle orientation, delamination and plasticity, it reports the ultimate daughter cell fate choice; (ii) it records progenitor/daughter cell locations—by maintaining spatial information of progenitor fate decisions, it allows for analysis of heterogeneity that may be driven by specific cell/cell and cell/niche interactions; and (iii) it is easily adoptable—by utilizing generic reagents and basic microscopy techniques, it can be readily assimilated into any research program.

Using this assay, we uncovered that Hras$^{G12V}$ can promote progenitor renewal or differentiation depending on whether it is active in a single cell or an oncogenic field. This surprising discovery led us to design a genetic screen that identified Ras effector Rassf5 as an oncogene-specific mediator of progenitor differentiation. Rassf5 has been long considered a tumor suppressor in epithelial cancers, where it is often silenced through methylation (*Donninger et al., 2016*) and was shown to promote both apoptosis and senescence (*Donninger et al., 2015*; *Khokhlatchev et al., 2002*). We demonstrate that Rassf5 suppresses progenitor cell renewal to limit oncogenic growth, which suggests a mechanism of Rassf5-mediated tumor suppression via differentiation. This model is consistent with recent evidence that increasing the rate of progenitor cell differentiation can suppress tumorigenic potential of known oncogenes (*Ying et al., 2018*), and that reducing the frequency of asymmetric spindle orientation can promote tumor development in skin (*Morrow et al., 2019*). How Rassf5 regulates progenitor fate choice is not known, but studies suggest a potential mechanism. Specifically, Rassf5 can activate Mst1, a driver of Hippo signaling (*Bitra et al., 2017*), which may lead to phosphorylation and eventual degradation of Yap. As Yap is enriched in the basal epidermal layer and acts to maintain it in an undifferentiated state (*Zhang et al., 2011*), Rassf5 may be a negative regulator of progenitor renewal through Hippo/Yap signaling.

We demonstrated that progenitor cells occupying the homogeneous environment of the inner clone were more likely to give rise to differentiated daughter cells, whereas the heterotypic environment on the edge of the clone appeared to permit progenitor renewal, as there was little difference between the renewal rate of single Hras$^{G12V}$ cells and late-stage edge cells. Changes in cell behavior influenced by homotypic/heterotypic cell interactions have been well documented (*Mishra et al., 2019*), and may be driven by relatively simple processes. For instance, the homotypic adhesions of Hras$^{G12V}$/Hras$^{G12V}$ inner cells could promote recruitment of more Ras and/or more Rassf5 to the

membrane, driving an increase in differentiation. Resultant interplay between the renewing edge and differentiating core of late-stage Hras$^{G12V}$ clones would then lead to oncogenic clone homeostasis.

In addition, our data show that WT cells near the Hras$^{G12V}$ clone were also more likely to differentiate. It is noteworthy that similar phenomenon was observed following clonal activation of PI3K, although in that context WT cells were more self-renewing and the oncogenic clone was in the process of being eliminated from the tissue through increased differentiation (*Ying et al., 2018*). Despite having opposite effects on WT cells, both observations suggest that epidermal progenitors can respond to environmental changes in order to promote the stable size of the progenitor compartment necessary for tissue maintenance. This is consistent with a model where regulation of stem cell fate choice, rather than being a sole product of cell autonomous signaling, is subject to progenitor cell dynamics among the nearest neighbors (*Mesa et al., 2018*). Of note, while that model was supported by observation that delamination of a basal cell precedes division of a neighboring progenitor in the context of adult tissue homeostasis (*Mesa et al., 2018*), it would be interesting to test if oncogenic activation of a potent mitogenic signal in a basal progenitor may reverse the relationship, and promote delamination of its neighbors.

In our model, biological outcome of a balanced clone is driven by inception of two progenitor populations with very different fates. As this emergence of distinct populations is defined by a progressive increase in clone size, which may be promoted by any oncogenic event that drives unbalanced proliferation, and overall round clone shape, which appears to be a general feature of epidermal organization and can therefore be viewed as a simple facilitator, it suggests that physical attributes of the clone can play a defining role. The dichotomy of growing edge and unstable core not only provides a model for how the growth of a single isolated clone can be constrained without requiring the presence of competing clones/mutations but may also explain the observation that oncogene-driven clones residing in aged epidermis grow to a relatively uniform size (*Martincorena et al., 2015*).

# Materials and methods

**Key resources table**

| Reagent type (species) or resource | Designation | Source or reference | Identifiers | Additional information |
|---|---|---|---|---|
| Genetic reagent (*Mus musculus*) | C57BL/6J | Jackson Laboratories | Stock #: 000664 | |
| Cell line (*Homo sapiens*) | 293-FT | Invitrogen | Cᴀᴛ #: R70007 | |
| Antibody | PerCP/Cyanine5.5 rat monoclonal anti-human/mouse CD49f | BioLegend | Cat #: 313617 | 1:50 |
| Antibody | Chicken polyclonal anti-GFP | Abcam | Cat #: ab13970 RRID:AB_300798 | IF 1:1000 |
| Antibody | Mouse monoclonal anti-BrdU | Invitrogen | Cat #: MoBU-1 | IF: 1:100 |
| Antibody | Mouse monoclonal anti-GAPDH | Proteintech | Cat #: 60004–1-Ig RRID:AB_2737588 | WB: 1:2000 |
| Antibody | Peroxidase AffiniPure goat polyclonal Anti-Mouse IgG (H+L) | Jackson Immunoresearch | Cat #: 115-035-003 | WB: 1:5000 |
| Antibody | Rabbit polyclonal anti-K10 | BioLegend | Cat #: Poly19054 | IF: 1:1000 |
| Antibody | Rabbit polyclonal anti-Nore1A | Donated by G. Clark | Cat #: PAS17071 | WB: 1:1000 |
| Antibody | Rabbit polyclonal anti-RFP | Rockland | Cat #: 6000-401-379 RRID:AB_11182807 | IF: 1:1000 |
| Antibody | Rat monoclonal anti-Nidogen | Santa Cruz Biotechnology | Cat #: sc-33706 RRID:AB_627519 | IF: 1:1000 |
| Commercial assay or kit | BCA Protein Assay Kit | Thermo Fisher | Cᴀᴛ #: 23225 | |

*Continued on next page*

*Continued*

| Reagent type (species) or resource | Designation | Source or reference | Identifiers | Additional information |
|---|---|---|---|---|
| Commercial assay or kit | Click-iT EdU cell proliferation kit for imaging, Alexa Fluor 555 | Thermo Fisher Scientific | Cᴀᴛ. #: C10337 | |
| Commercial assay or kit | DNeasy Blood and Tissue Kit | Qiagen | Cat #: 69504 | |
| Commercial assay or kit | iScript Reverse Transcription Supermix | Bio-Rad | Cat #: 1708840 | |
| Commercial assay or kit | Power SYBR Green PCR Master Mix | Thermo Fisher Scientific | Cat. #: 4367660 | |
| Commercial assay or kit | SuperSignal West Femto Maximum Sensitivity Substrate | Thermo Fisher | Cᴀᴛ #: 34094 | |
| Commercial assay or kit | Zero Blunt TOPO kit | Invitrogen | Cᴀᴛ #: 450245 | |
| Recombinant DNA reagent | FLAG-Nore1 | Addgene | Cat #: 1975 RRID:Addgene_1975 | |
| Recombinant DNA reagent | tdTomato-C1 | Addgene | Cat #: 54653 RRID:Addgene_54653 | |
| Software, algorithm | Prism | GraphPad | RRID:SCR_002798 | |
| Software, algorithm | Zen Black | Zeiss | RRID:SCR_018163 | |
| Other | M.O.M. buffer | Vector Labs | Cat #: BMK-220 | |

## Animals

All mice were on a C57BL/6 or a C56BL/6-Tyr$^{c-2J}$ background including, *Hras*$^{G12V/G12V}$ (*Chen et al., 2009*), *Tg(K14-cre)1Efu* (*Vasioukhin et al., 1999*), *Gt(Rosa)26Sor*$^{tm1(eYFP)Cos/+}$ (Jackson Laboratories), and *Gt(ROSA)26Sor*$^{tm4(ACTB-tdTomato,-EGFO)Luo}$ (Jackson Laboratories). Female and male animals were used in equal numbers. Randomization and blinding were not used in this study.

## CFI assay

EdU was administered via intraperitoneal injection to adult animals followed 2 hr later by BrdU. Animals were euthanized 24 hr after EdU injection. Head skin was removed and samples were incubated in 30 mM EDTA overnight at 4°C and then for 2 hr at 37°C to separate dermis from epidermis. The epidermis was then gently separated from the dermis by scrapping with scalpel, fixed for 1 hr in 4% PFA, followed by incubations for 1 hr in 0.5% Triton and then 1 hr in blocking buffer containing 0.5% Triton. Next, tissues were incubated with primary chicken anti-GFP antibody overnight at 4°C (to label transduced clones), and then processed for EdU expression using Click-iT technology (Invitrogen) according to the manufacturer's instructions. This was followed by a 15-min treatment in 2N HCL at 37°C to denature DNA, and two washes in 0.1 M sodium borate, pH 8.5. Following this quenching step, tissues were incubated in anti-chicken secondary (Invitrogen) for 1 hr. After incubation in M.O.M. buffer (Vector Labs) for 1 hr, tissues were processed for BrdU detection, stained with 4,6-diamidino-2-phenylindole (DAPI; Life Technologies), and mounted in ProLong Gold (Invitrogen). Images were collected using a Zeiss LSM700 system with a Plan-Apochromat 40X/1.4 oil objective. In order to create images with enough detail to accurately score division type, Z-stacks, spanning the full thickness of the epidermis, with 1 μm slice intervals were generated. Fates were scored by determining daughter cell location (basal or suprabasal layer) and morphology (cuboidal or squamous).

## EdU-BrdU pulse-chase differentiation assay and renewal rate quantification

E18.5 tissues were embedded in OCT. After sectioning, tissues were fixed in 4% PFA, washed in 0.1% Triton followed by incubation for 1 hr at room temperature in blocking buffer. Tissues were processed for K10 detection, and then processed for EdU expression using Click-iT technology (Invitrogen) according to the manufacturer's instructions. This was followed by incubation in 2N HCL for

30 min at 37°C to denature DNA and two washes with 0.1 M sodium borate, pH 8.5 for 15 min. Tissue sections were then incubated in M.O.M. buffer (Vector Labs) for 1 hr and processed for BrdU detection followed by mounting in ProLong Gold (Invitrogen). Rate of renewal was calculated as the proportion of EdU+/K10- cells out of all EdU+ cells. More detailed explanations of the methodology can be found elsewhere (*Ying et al., 2018*).

### Proliferation assay
Head skin was processed using the CFI assay protocol, described above. To determine proliferation rate, we calculated the proportion of EdU+ progenitor cells within transduced clones.

### Immunofluorescence
The following primary antibodies were used: chicken anti-GFP (ab13970, 1:1000; Abcam); mouse anti-BrdU (MoBU-1, 1:100; Invitrogen); rabbit anti-K10 (Poly19054, 1:1,000 BioLegend); rabbit anti-RFP (6000-401-379, 1:1000; Rockland); rat anti-Nidogen (sc-33706, 1:1000; Santa Cruz Biotechnology). Tissues were processed for immunostaining as previously described[4,5] and mounted in Pro-Long Gold (Invitrogen) with or without 4,6-diamidino-2-phenylindole (DAPI; Life Technologies). Confocal images were taken on a Zeiss LSM700 system using a Plan-Apochromat 40X/1.4 oil objective. Image processing was done using Zeiss Zen and ImageJ software.

### Cell packing quantification
Head skin was prepared for whole-mount confocal imaging. Images were collected using a Zeiss LSM700 system with a Plan-Apochromat 40X/1.4 oil objective. Clone area was measured using ImageJ. Cell density was calculated using the formula: clone density = (# basal cells / clone area).

### Clone morphology and circularity quantification
Head skin whole-mounts were imaged on a Zeiss LSM700 system with a Plan-Apochromat 40X/1.4 oil objective. Using ImageJ clone outlines were traced and an ellipse was fitted to each clone. The perimeter and area of the ellipse was measured using ImageJ. Circularity was calculated using the following equation: circularity = $4\pi(\text{area} / \text{perimeter}^2)$.

### Western blot
Head skin from E18.5 animals was incubated in 2 mg/mL dispase at 37°C for 1 hr to separate dermis from epidermis. The epidermis was then digested in RIPA buffer with phosphatase and protease inhibitor cocktails (Santa Cruz Biotechnology) for 30 min on ice followed by sonication. Supernatants were assayed for protein concentration using the Pierce BCA Protein Assay Kit (Thermo Fisher). Western blotting was performed using a Novex system (Invitrogen). Membranes were incubated with primary antibody overnight and then incubated with HRP-conjugated secondary antibodies (Jackson Immunoresearch) for 1 hr at room temperature. Membranes were developed using Super-Signal West Femto Maximum Sensitivity Substrate (Thermo Fisher). Chemiluminescent signals were detected using an Odyssey Fc system (LI-COR). The following primary antibodies were used: rabbit anti-Nore1A (PAS17071, 1:1000; donated by G. Clark), mouse anti-GAPDH (60004–1-Ig, 1:2000; Proteintech).

### In vivo genetic screens
Head skin of E18.5 mice was collected and digested in 2 mg/mL dispase at 37°C for 1 hr to separate dermis from epidermis. Epidermal tissue was then digested in 0.25% trypsin for 30 min to isolate single cells. Cells were labeled with CD49f/$\alpha_6$-integrin-PerCP (1:50; BioLegend) and purified using fluorescence activated cell sorting (FACS) using BD FACSAria II (BD Biosciences). Genomic DNA was extracted using DNeasy Blood and Tissue Kit (Qiagen). Barcode preamplification, sequencing and data processing using the Deseq2 program were performed as previously described (*Ying et al., 2018*).

### Lentiviral constructs
A mouse Rassf5 expression construct (Addgene) was cloned into tdTomato-C1 vector (Addgene) at EcoRI/SmaI sites. Rassf5-tdT was then cloned into a modified pLX Cre EF1 vector[3] using the Zero

Blunt TOPO kit (Invitrogen). To make inducible Rassf5 and CreER constructs, Rassf5-tdT was cloned into a pLKO2 TRE vector at NsiI/BspEI sites. CreER was cloned into a pLKO2 rtTA construct at SalI/NheI sites.RNA interference-mediated gene depletion was achieved using pLKO1 shRNA vectors from the mouse TRC1.0 shRNA library (Sigma-Aldrich). pLKO-Cre vectors were used to generate Cre-shRNA expression constructs (*Beronja et al., 2013*; *Beronja et al., 2010*).

### Lentiviral production and transduction

Large-scale production and concentration of lentivirus was performed as previously described (*Beronja et al., 2010*). Detailed descriptions of in vitro and in vivo lentiviral transductions and in-utero-guided lentiviral transduction can be found elsewhere (*Beronja et al., 2013*; *Beronja et al., 2010*).

### mRNA quantifications

Total RNA was isolated from cultured keratinocytes or from E18.5 FACS-sorted epidermal cells using RNeasy Plus Mini Kit (Qiagen). Complementary DNA was generated from 1 μg of total RNA using the iScript Reverse Transcription Supermix (Bio-Rad). Quantitative PCR was performed with SYBR Green PCR Master Mix (Thermo Fisher) and with gene-specific and *Rpl16* control primers.

### Intravital imaging using two-photon microscopy

An LSM 780 multiphoton, laser scanning confocal microscope (Zeiss) was used in intravital imaging. Details of the technique have been previously described (*Rompolas et al., 2013*). Mice were anesthetized and further immobilized using a custom device so that the head skin could be imaged without interfering vibrations. GFP and Tomato signals were captured using a 940 nm laser.

### Statistical information

All experiments were performed at least in triplicate, and all quantitative data are expressed as mean ± s.d. Differences between conditions were analyzed in Prism 7 (GraphPad) using Student's t-test. Significant differences were considered when $p < 0.05$.

## Acknowledgements

We thank J Fagin for sharing the inducible *Hras*$^{G12V}$ mouse; G Clark for the Rassf5 antibodies; Comparative Medicine (AAALAC accredited; G Roble, director) for care of mice in accordance with National Institutes of Health (NIH) guidelines; Genomics (J Delrow, director) for sequencing; Scientific Imaging (J Vazquez) for advice; Flow Cytometry (A Berger, director) for flow cytometry and FACS. This research was funded in part through the NIH/NCI Cancer Center Support Grant P30 CA015704, NIH/NIAMS R01-AR070780 (SB), and Cell and Molecular Biology Training Grant (MS)

## Additional information

### Funding

| Funder | Grant reference number | Author |
| --- | --- | --- |
| National Institute of Arthritis and Musculoskeletal and Skin Diseases | AR070780 | Slobodan Beronja |
| Cell and Molecular Biology Training Grant | Graduate Student Fellowship | Madeline Sandoval |
| National Cancer Institute | P30 CA015704 | Slobodan Beronja |

The funders had no role in study design, data collection and interpretation, or the decision to submit the work for publication.

## Author contributions
Madeline Sandoval, Conceptualization, Data curation, Formal analysis, Investigation, Methodology, Writing - original draft, Writing - review and editing; Zhe Ying, Investigation; Slobodan Beronja, Conceptualization, Supervision, Funding acquisition, Writing - original draft, Writing - review and editing

## Author ORCIDs
Slobodan Beronja (iD) https://orcid.org/0000-0002-6769-9261

## Ethics
Animal experimentation: Animal experimentation: Mice were housed and cared for in an AAALAC-accredited facility at Fred Hutchinson Cancer Research Center. All animal experiments were conducted under approved IACUC protocol number 50814 (approval date 12/01/2018-11/29/2021).

## Decision letter and Author response
Decision letter https://doi.org/10.7554/eLife.54618.sa1
Author response https://doi.org/10.7554/eLife.54618.sa2

# Additional files

## Supplementary files
• Source data 1. Quantification and statistical analyses of shRNA enrichment and depletion in progenitor and differentiated cells of wild type and Hras$^{G12V}$ epidermis. Relative enrichment of each shRNA targeting putative regulators of progenitor renewal in epidermal basal progenitors relative to differentiated suprabasal cells, as quantified by DESeq2 analyses.

• Supplementary file 1. Regulators of progenitor renewal in wild type and Hras$^{G12V}$ epidermis. Candidates regulators of progenitor renewal as identified by DESeq2 analyses of shRNA enrichment in epidermal basal progenitors relative to differentiated suprabasal cells. Positive fold-change indicates shRNA enrichment in basal progenitors.

• Transparent reporting form

## Data availability
All data generated or analysed during this study are included in the manuscript and supporting files. Source data file has been provided for Supplementary File 1.

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
