## [Decision Letter]

**Acceptance summary:**

The work addresses an important question, namely what happens when cells acquire an oncogene that will ultimately progress to cancer. The authors nicely show that tumor cells in a stratified mouse epidermis generates two populations of oncogenic cells: ones at the tumor border that favor self-renewal and ones in the center of the clone that favor differentiation. As the clone grows, the intra-clonal competition between the two populations results in stabilized oncogenic growth. This knowledge could be widely applicable to many cancers derived from stratified tissues.

**Decision letter after peer review:**

Thank you for submitting your article "Interplay of opposing fate choices stalls oncogenic growth in skin epithelium" for consideration by *eLife*. Your article has been reviewed by three peer reviewers, one of whom is a member of our Board of Reviewing Editors, and the evaluation has been overseen by Richard White as the Senior Editor. The following individual involved in review of your submission has agreed to reveal their identity: Scott Williams (Reviewer #2).

The reviewers have discussed the reviews with one another and the Reviewing Editor has drafted this decision to help you prepare a revised submission.

Summary:

In this work the authors have created a genetically engineered mouse model where the effects of activating HRAS in the epithelium allows for the analysis of cell turnover/division at single cell resolution. They use this elegant experimental model to discover that the growth of oncogenic lesions is nuanced with both pro-proliferative and restraining forces at play. They also demonstrate that this model can be used for genetic screens and performed a screen that identified Rassf5 as a negative regulator of Hras.

Essential revisions:

1) The description of the CFI assay is overly complicated and given it is really just a slight amendment from the assay they previously employed in Ying, Sandoval and Beronja, 2018, the novelty of this assay is a little bit oversold, especially in the Discussion. The labelling of Figure 2B is also quite confusing – the EdU pulse is actually 2.5 hours in the middle panel of images, and the top panel is tissue that hasn't been subjected to a BrdU injection if I am understanding correctly but this isn't clear initially. Also what the %s are referring to in terms of either total cell population of just doublets is also unclear. Also the CFI method relies on the fact that a cell must divide (incorporate EdU or BrdU) in order to be labeled and counted. A cell that delaminates, without having recently dividing, would not be considered. Since there is evidence that delamination precedes division in the adult epidermis (Mesa et al., 2018), it seems that the concept of delamination should at least be considered (and this paper cited in the Discussion).

2) By way of comparison, none of the analyses in Figure 3A are performed for WT clones. Does inter-clone heterogeneity and the tendency toward clonal circularity exist in WT clones? It would be nice to have some of these analysis performed in WT clones to confirm that the Hras^G12V^ behaviors are unique. In particular, single cell sequencing might be quite illuminating here.

3) In Figure 3G, although they do see significantly more self-renewal on the edges of the clone, there is not significantly more differentiation in the centre. Although the trend is there, the authors should be careful with their conclusions and this should be discussed. It seems like the main difference that accounts for what they see is the changes in self-renewal. In addition, do WT clones that are in contact with the Hras^G12V^ edge cells behave any differently than WT clones far removed from a Hras^G12V^ clone? The notion of competition between WT and mutant clones discussed by this group (subsection “Emerging heterogeneity in expanding Hras^G12V^ clones impacts progenitor renewal behavior) and others suggests that these WT "edge" cells may have unique properties.

4) The authors claim in Figure 1—figure supplement 1 and in the subsection “Expression of activated Hras^G12V^ promotes progenitor cell renewal in adult skin epithelium”, that neither apoptosis nor cell cycle length are significantly different between WT and Hras^G12V^ mutants; thus, they exclude these factors from their growth model (Figure 1G). However, these analyses were performed in embryonic back rather than adult head epidermis, and presumably in tissue that has ubiquitous rather than clonal induction of Hras^G12V^. Since the authors show in Figure 4A that there are no apparent differences in progenitor renewal rates between WT and Hras^G12V^ in embryonic clones, these assumptions may not be valid for their model. The authors should at a minimum openly disclose this caveat. Also, additional details on how the growth models (Figures 1G, 2K) were built should be included in the Materials and methods.

5) There is a general lack of n values for the majority of quantified data. The number of animals analyzed are listed but not the number of data points collected. These should be added to figures and/or legends. In addition, statistics are missing from Figure 2G-J.

6) In order to properly contextualize the results shown in Figure 3, it would be important to show that in a large field of Hras-expressing cells (i.e. K14-Cre driven or high titer lentivirus driven expression), we don't see clonal shapes trending towards being round. This critical control would serve to reinforce the idea that in the case of clonal Hras expression, it is really interactions between wild-type and oncogenic cells that eventually contributes to the restraint of overgrowth in the tissue. An analysis of clonal growth in WT tissue would also serve to illustrate and further reinforce the same point.

7) The screen was quite limited and although they identified a new regulator, this was selected from candidate genes. How large a screen can be performed? In addition, no mention or discussion is made of the screen hits that were enriched in the SB population in Hras^G12V^ tissue, although there appear to be several. Also, it should be clarified which genetic backgrounds were used for the screen. I assume this is the Krt14-Cre; Hrasll-G12V/+, and that WT would be without either the Krt14-Cre or Hras^G12V^ allele, but this should be explicitly stated in the text. Data regarding the enrichment/depletion of their screen hits (Figure 4E, F) should be provided.

8) The observations regarding intraclonal heterogeneity are very interesting and quite well described. However, that Rassf5 is a bona fide contributor to these spatially distinct dynamics is not sufficiently convincing. More thorough analysis should be included to suggest that there are changes to these dynamics when Rassf5 expression is perturbed, either by shRNA-mediated knockdown, or by over-expression. It is puzzling that no data is shown to support this, and in both the initial description of these data in the Results, and in the Discussion, the authors mention that in other systems Rassf5 is thought to restrain oncogenic Ras activity through induction of senescence of apoptosis. However, according to their data, there is neither senescence nor apoptosis induced in the case of the epidermis (although this data could also be more convincing). At the very least, analysis similar to what is shown in Figure 3 should be included in the case of shRassf5 and/or Rassf5 overexpression to convincingly demonstrate that the phenomenon they describe in the first three figures can indeed be linked to Rassf5 function. Simply saying that the clones are more triangular upon Rassf5 overexpression is not sufficient. What do these clones look like in whole mount? Are they still circular like the Hras^G12V^ clones, or is this lost? Is the heterogeneity of cell behaviors between the "inner" and "edge" cells maintained or lost? Is proliferation or apoptosis affected? Also, the Rassf5 knockdown and over-expression are assessed at E18.5 rather than later post-natal time points which makes it furthermore difficult to include the relevance between the phenotypes and what they describe earlier in the paper. Another potential experiment to include that would help their case would be to stain for Rassf5 to see if there are differences on the protein expression level within clones – if there is a working antibody available this should be included.

---

## [Author Response]

Essential revisions:1) The description of the CFI assay is overly complicated and given it is really just a slight amendment from the assay they previously employed in Ying, Sandoval and Beronja, 2018), the novelty of this assay is a little bit oversold, especially in the Discussion. The labelling of Figure 2B is also quite confusing – the EdU pulse is actually 2.5 hours in the middle panel of images, and the top panel is tissue that hasn't been subjected to a BrdU injection if I am understanding correctly but this isn't clear initially. Also what the %s are referring to in terms of either total cell population of just doublets is also unclear. Also the CFI method relies on the fact that a cell must divide (incorporate EdU or BrdU) in order to be labeled and counted. A cell that delaminates, without having recently dividing, would not be considered. Since there is evidence that delamination precedes division in the adult epidermis (Mesa et al., 2018), it seems that the concept of delamination should at least be considered (and this paper cited in the Discussion).

We appreciate the reviewers’ comments on the CFI assay and the desire for more clarity. As the reviewers noted, it is adapted from the assay we used in Ying, Sandoval and Beronja, 2018, and we now clearly state so in both the Results and Discussion sections. However, we spent considerable time modifying and validating the initial assay into what is now the CFI assay. Most importantly, CFI assay can answer questions that we could not address using our previous method (described in the Discussion). Because of this, we believe that the CFI assay holds merit. As suggested, we made a series of modifications to Figure 2B, both to correct issues brought up by the reviewers and give the reader a better understanding of the assay.

The reviewers also raised an important point regarding CFI assay’s dependence on incorporation of EdU/BrdU. Indeed, as the CFI assay depends on nucleotide labeling of daughter progenitors, any division/delamination event (i.e. renewal/differentiation fate choice), made by an epidermal progenitor derived from a mother cell that went through its S phase outside of the 2hr EdU pulse, is not scored by the CFI. As the assay reports the *frequency* of particular fate choice and uses that to calculate the *rate* of renewal, these prior events do not impact either statistic (similar to, for instance, the rate of apoptosis based on a snapshot of Casp3 expression). The CFI scores daughter cell fates within a 24hr window, which brings up an additional question, of whether this is sufficient time to account for cellular fates that labeled progenitors ultimately make. Specifically, if daughter cell delamination occurs with any significant frequency outside of the 24hr window we employ, these delamination events would be missed. As a result, our assay would be expected to over-report the tissue renewal rate. There are several observations that suggest that this is not the case. In previous work we showed that the number of newborn daughter cells with strong expression of suprabasal marker K10, detected by IF, plateaus by ~6hrs following EdU pulse, and that these K10 high progenitors never divide again (Ying, Sandoval and Beronja, 2018). In addition, studies by the Greco and our labs, which employed direct intravital imaging of photo-labeled progenitors (i.e. an assay independent of progenitor labeling based on nucleotide incorporation), and progenitor cell fate scoring based not on their location in the epidermal basal layer after a 24hr period but by their ability to divide again (a hallmark of progenitor potential) arrived at the adult tissue renewal rates of 0.50±0.04 (calculated from quantification in Figure 2A; Rompolas et al., 2016 Science) and 0.51±0.01 (Ying, Sandoval and Beronja, 2018). This is not significantly different from the renewal rate of 0.50±0.01, which we report for the wild type tissue in the current study, and is further consistent with overall neutral growth and renewal rate required for long-term tissue maintenance. Nevertheless, in the revised manuscript we highlight that CFI employs a 24hr snapshot to assess progenitor cell fate choice, thus leaving space for possibilities that delaminated cells can maintain their mitotic potential, or that basal progenitors delaminate with any significant frequency after the 24hr period.

Lastly, we fully agree with the reviewer’s suggestion to discuss Mesa et al., especially as their observation of local regulation of stem cell fate choice (i.e. basal cell delamination is accompanied by division of a neighboring progenitor) may contribute to the phenomenon we observe where increased renewal along the edge of the Hras^G12V^ clone is accompanied by reduced renewal of neighboring wild type cells. Our revised manuscript now has an entire paragraph dedicated to discussing our observations in the context of their elegant model.

2) By way of comparison, none of the analyses in Figure 3A are performed for WT clones. Does inter-clone heterogeneity and the tendency toward clonal circularity exist in WT clones? It would be nice to have some of these analysis performed in WT clones to confirm that the Hras^G12V^ behaviors are unique. In particular, single cell sequencing might be quite illuminating here.

The reviewers noted that analysis of inter-clone heterogeneity of Figure 3 was not done on WT clones. In response, we performed a series of experiments in WT epidermis.

On the point of clonal circularity, we provide data that small WT clones are also circular in shape (Figure 3—figure supplement 1A, B). While we could not compare circularity for WT and Hras^G12V^ clones of the same size (see below), this experiment suggests that clone growth in the epidermis inherently trends towards a circular morphology. Importantly, we want to clarify that our model does not require that circularity to be driven by Hras^G12V^, but that its natural existence is critical for the emergence of two populations of cells (outer/inner) and the dynamic in their relative contribution that favors outer cells early on but inner cells over a longer time span. In our revised Discussion we clarify this view of circularity as a naturally-occurring yet critical facilitator of dynamics of clone composition.

In our revised manuscript we also show that WT clones initiated from a single progenitor do not grow to a size sufficient for emergence of inner/outer cell populations (Figure 3—figure supplement 1C). To nevertheless address the reviewers’ comment we conducted an additional experiment using larger WT fields derived from several neighboring progenitors. We found that the division choices of inner and outer cells in that context were not significantly different, and further that their overall renewal rates were not significantly different from that of small WT clones (Figure 3—figure supplement 1D, E). These analyses suggest that the inter-clone heterogeneity is independent of lentiviral transduction, expression of Cre recombinase and/or GFP, and is specific to Hras^G12V^-expression.

3) In Figure 3G, although they do see significantly more self-renewal on the edges of the clone, there is not significantly more differentiation in the centre. Although the trend is there, the authors should be careful with their conclusions and this should be discussed. It seems like the main difference that accounts for what they see is the changes in self-renewal. In addition, do WT clones that are in contact with the Hras^G12V^ edge cells behave any differently than WT clones far removed from a Hras^G12V^ clone? The notion of competition between WT and mutant clones discussed by this group (subsection “Emerging heterogeneity in expanding Hras^G12V^ clones impacts progenitor renewal behavior) and others suggests that these WT "edge" cells may have unique properties.

We agree with both points raised by our reviewers. In an effort to further probe the trend that we saw in data shown in old Figure 3G, we increased the n-value by an additional set of 24 clones.

Our new analysis shows a statistically significant increase in differentiation in the center of Hras^G12V^ clones, supporting the model where the edge cells are undergoing more renewing divisions while the inner portion of the clone is pro-differentiation (Figure 3G, H).

To address the question about the behavior of WT “edge” cells, we analyzed the behavior of WT cells adjacent to an Hras^G12V^ clone and WT cells further away (>150 µm) from an Hras^G12V^ clone. We found that symmetric renewal divisions occurred less often in WT cells adjacent to an Hras^G12V^ clone (Figure 3I, J), suggesting that WT cells are responding to the local environment and compensate for the pro-renewal edge cells of Hras^G12V^ clones. We highlight this observation in our revised Discussion, and relate it to published work from the Greco lab (Mesa et al., 2018) and our own (Ying, Sandoval and Beronja, 2018).

4) The authors claim in Figure 1—figure supplement 1 and in the subsection “Expression of activated Hras^G12V^ promotes progenitor cell renewal in adult skin epithelium”, that neither apoptosis nor cell cycle length are significantly different between WT and Hras^G12V^ mutants; thus, they exclude these factors from their growth model (Figure 1G). However, these analyses were performed in embryonic back rather than adult head epidermis, and presumably in tissue that has ubiquitous rather than clonal induction of Hras^G12V^. Since the authors show in Figure 4A that there are no apparent differences in progenitor renewal rates between WT and Hras^G12V^ in embryonic clones, these assumptions may not be valid for their model. The authors should at a minimum openly disclose this caveat. Also, additional details on how the growth models (Figures 1G, 2K) were built should be included in the Materials and methods.

The reviewers raise the fair point of limitations of using apoptosis/senescence data generated in embryonic Hras^G12V^ tissue in a model of adult tissue growth. For the revised manuscript we conducted analyses of apoptosis and senescence in WT and Hras^G12V^ clones induced in the adult head epidermis (Figure 1—figure supplement 1A, B). They show that there are no significant changes in apoptosis rates between adult WT and Hras^G12V^ clones, and further that there is no evidence for senescence. We provide these new analyses and note that we have decided to remove the mathematical model from the study at the suggestion of the reviewers.

5) There is a general lack of n values for the majority of quantified data. The number of animals analyzed are listed but not the number of data points collected. These should be added to figures and/or legends. In addition, statistics are missing from Figure 2G-J.

Thank you for pointing this out. We have rectified these oversights and now provide details on sample size in each figure legend and the statistical analysis for Figure 2F-K.

6) In order to properly contextualize the results shown in Figure 3, it would be important to show that in a large field of Hras-expressing cells (i.e. K14-Cre driven or high titer lentivirus driven expression), we don't see clonal shapes trending towards being round. This critical control would serve to reinforce the idea that in the case of clonal Hras expression, it is really interactions between wild-type and oncogenic cells that eventually contributes to the restraint of overgrowth in the tissue. An analysis of clonal growth in WT tissue would also serve to illustrate and further reinforce the same point.

The reviewers have raised an interesting point about Hras^G12V^ inducing circularly shaped clones. In considering this idea, we would like to clarify that Ras is not driving roundness but that the overall tissue architecture induces a single cell to grow into a clone that has a circular shape.

Indeed, our analysis of WT clone shape over 24 weeks suggests that WT clones also sustain a round shape (Figure 3—figure supplement 1A, B). This suggests that the native tissue architecture plays a role in impeding Hras growth. Furthermore, we wish to clarify that it is not "interactions between WT and oncogenic cells that eventually contribute to the restraint of overgrowth in the tissue”. Hras^G12V^ clones are self-limiting because the inner core of cells is pro-differentiation (Figure 3H). Indeed, neighboring WT cells respond to the Hras^G12V^ clone by undergoing divisions that produce differentiated daughter cells (Figure 3I, J), indicating that this population is not inhibiting the growth of the Hras^G12V^ clone but is instead reacting to the pro-renewal fate choices of the Hras^G12V^ edge cells (Figure 3K).

7) The screen was quite limited and although they identified a new regulator, this was selected from candidate genes. How large a screen can be performed? In addition, no mention or discussion is made of the screen hits that were enriched in the SB population in Hras^G12V^ tissue, although there appear to be several. Also, it should be clarified which genetic backgrounds were used for the screen. I assume this is the Krt14-Cre; Hrasll-G12V/+, and that WT would be without either the Krt14-Cre or Hras^G12V^ allele, but this should be explicitly stated in the text. Data regarding the enrichment/depletion of their screen hits (Figure 4E, F) should be provided.

The scope of the type of screen we conducted can range from a handful of candidates (e.g. bona fide cancer drivers in Ying, Sandoval and Beronja, 2018), to several hundred (e.g. putative cancer drivers in Ying and Beronja, 2020 Cell Stem Cell), to the whole genome (e.g. Beronja et al., 2013). In general, we tend to use large screens in the hypothesis building stages of a study, and biologically focused ones when addressing a specific hypothesis. Here, our screen was focused on known Ras effectors to test the hypothesis that increased differentiation in oncogenic clones was caused by Hras^G12V^ expression and mediated by one of its effectors.

The reviewers’ comments on how to improve the discussion of our in vivo genetic screen is appreciated. We now explicitly state the genetic backgrounds of the animals we used. In addition, we include a table that provides details of enrichment/depletion scores for all genes tested in the screen (Supplementary file 1). Lastly, we highlight Ras effectors that were enriched in the suprabasal population (i.e. Rgl2, Rgs14, Rassf4), in revised text and revised Figure 4.

8) The observations regarding intraclonal heterogeneity are very interesting and quite well described. However, that Rassf5 is a bona fide contributor to these spatially distinct dynamics is not sufficiently convincing. More thorough analysis should be included to suggest that there are changes to these dynamics when Rassf5 expression is perturbed, either by shRNA-mediated knockdown, or by over-expression. It is puzzling that no data is shown to support this, and in both the initial description of these data in the Results, and in the Discussion, the authors mention that in other systems Rassf5 is thought to restrain oncogenic Ras activity through induction of senescence of apoptosis. However, according to their data, there is neither senescence nor apoptosis induced in the case of the epidermis (although this data could also be more convincing). At the very least, analysis similar to what is shown in Figure 3 should be included in the case of shRassf5 and/or Rassf5 overexpression to convincingly demonstrate that the phenomenon they describe in the first three figures can indeed be linked to Rassf5 function. Simply saying that the clones are more triangular upon Rassf5 overexpression is not sufficient. What do these clones look like in whole mount? Are they still circular like the Hras^G12V^ clones, or is this lost? Is the heterogeneity of cell behaviors between the "inner" and "edge" cells maintained or lost? Is proliferation or apoptosis affected? Also, the Rassf5 knockdown and over-expression are assessed at E18.5 rather than later post-natal time points which makes it furthermore difficult to include the relevance between the phenotypes and what they describe earlier in the paper. Another potential experiment to include that would help their case would be to stain for Rassf5 to see if there are differences on the protein expression level within clones – if there is a working antibody available this should be included.

The reviewers raise a fair point on the importance of including additional functional analyses of Rassf5 in the adult tissue, and in the context of its previously described role in regulation of apoptosis and senescence. We performed apoptosis and senescence assays in adult Hras^G12V^ epidermis transduced with two different shRassf5 constructs, and found neither process to be significantly changed when compared to Hras^G12V^ control (Figure 5—figure supplement 1). Focusing on Rassf5-mediated regulation of progenitor renewal we demonstrated for embryonic tissues, we now show that Rassf5 knockdown in adult Hras^G12V^ epidermal clones erased the difference in progenitor renewal rates between inner and outer clone compartments (Figure 5J), and resulted in an overall increase in population progenitor renewal rate (Figure 5K). This indicates that Rassf5 is a regulator of progenitor renewal in the adult epidermis. Furthermore, to complement the loss-of-function analysis of Rassf5, we designed a two component Doxinducible system to activate Hras^G12V^ in adult epidermis, and follow it by overexpression of Rassf5 (Figure 5L, M). Here, our analyses show that Rassf5 expression biased early clone composition towards a differentiated fate to the point where basal progenitors were rapidly lost, suggesting that Rassf5 is sufficient for increased differentiation in the adult epidermis (Figure 5N, O). On the last point, we followed the reviewer’s suggestion to use immunostaining on clones. After testing 4 commercially available antibodies and 2 donated by other labs, and multiple fixation/staining protocols on wild type and Rassf5 knockdown and overexpression clones, we could not find an antibody that worked in tissue immunofluorescence or immunohistochemistry with any specificity.